# The Role of Mu Opioid Receptors in High Fat Diet-Induced Reward and Potentiation of the Rewarding Effect of Oxycodone [note 1]

**DOI:** 10.3390/life13030619

**Published:** 2023-02-23

**Authors:** Asif Iqbal, Abdul Hamid, Syed Muzzammil Ahmad, Kabirullah Lutfy

**Affiliations:** Department of Pharmaceutical Sciences, College of Pharmacy, Western University of Health Sciences, Pomona, CA 91766-1854, USA

**Keywords:** high fat diet, conditioned place preference, mu opioid receptor, knockout mice, endogenous opioid system, oxycodone

## Abstract

Excessive high fat diet (HFD) consumption can induce food addiction, which is believed to involve the communication between the hypothalamus and mesolimbic dopaminergic neurons, originating in the ventral tegmental area (VTA) and projecting to the nucleus accumbens (NAc). These brain areas are densely populated with opioid receptors, raising the possibility that these receptors, and particularly mu opioid receptors (MORs), are involved in rewards elicited by palatable food. This study sought to investigate the involvement of MORs in HFD-induced reward and if there is any difference between male and female subjects in this response. We also assessed if exposure to HFD would alter the rewarding action of oxycodone, a relatively selective MOR agonist. The place conditioning paradigm was used as an animal model of reward to determine if short-time (STC, 2 h) or long-time (LTC, 16 h) conditioning with HFD induces reward or alters the rewarding action of oxycodone. Male and female C57BL/6J mice as well as MOR knockout and their wildtype littermates of both sexes were tested for basal place preference on day 1 and then conditioned with an HFD in one chamber and a regular chow diet (RCD) in another chamber for 2 h on alternate days. Three sets of STC were used, followed by a set of LTC. Each set of conditioning consisted of two conditioning with RCD and two conditioning with HFD. Mice were tested for place preference after each set of STC and again after LTC. Controls were conditioned with RCD in both conditioning chambers. Following the last place preference test, mice were treated with oxycodone and conditioned in the HFD-paired chamber and with saline in the RCD-paired chamber for one hour once a day to explore the possibility if the HFD could alter oxycodone reward. The result showed that HFD induced conditioned place preference (CPP) in male but not female subjects. However, oxycodone conditioning elicited reward in both male and female mice of the HFD group but not the control group, showing that prior conditioning with HFD potentiated the rewarding action of oxycodone. The latter response was mediated via MORs, as it was blunted in MOR knockout mice. Similarly, HFD-induced CPP was blunted in male MOR knockout mice, suggesting sexual dimorphism in this response.

## 1. Introduction

An increase or decrease in food consumption is regulated by several orexigenic or anorexigenic neurons located in the central nervous system (CNS). These neurons are activated or deactivated through signals elicited by hormones secreted from the CNS or periphery, such as the stomach, intestine, and pancreas. These neurons also communicate with the ventral tegmental area (VTA) via the paraventricular nucleus of the hypothalamus (PVN). The dopaminergic neurons in the VTA project to the nucleus accumbens (NAc) [1], a brain area often referred to as the hedonic hotspot [2]. The intake of palatable food beyond the energy requirement activates the reward circuit, causing dopamine release in the NAc [3], thereby leading to binge eating and possibly food addiction. There is also evidence showing alterations in mesolimbic dopaminergic neurons in obese individuals, as observed with addictive drugs (for reviews, see [4,5]). There is also evidence showing that animals exposed to HFD express enhanced cocaine reward [6], but blunted ethanol preference [7]. However, it is unclear whether HFD consumption is associated with reward. Thus, one of the goals of this study was to assess if conditioning with HFD would induce reward.

Both palatable diet and addictive drugs activate the mesolimbic dopaminergic system (for reviews, see [5,8]). The endogenous opioid system, consists of endogenous opioids and their receptors, has been implicated in feeding and diet-induced obesity [9]. In this regard, palatable food exposure has been shown to increase MOR expression among obese rats [10]. Similarly, excessive HFD consumption has been implicated in dysregulated dopamine and opioid gene expression [11]. In addition, the overeating of palatable food has been shown to involve the endogenous opioid system [12]. Moreover, consumption of either palatable or non-palatable/regular food has been reported to induce the release of endogenous opioid peptides [13]. Additionally, local administration of MOR antagonists, such as naltrexone, in the NAc and amygdala has been reported to reduce palatable diet intake [14,15]. Overall, these observations suggest that endogenous opioid peptides can alter food consumption and may play a functional role in food reward. Thus, in the current research, we have combined the place conditioning paradigm with food intake to assess if conditioning with HFD induces rewards in mice and if the duration of conditioning with HFD would be important in the acquisition of a conditioned place preference (CPP). Considering that MOR is implicated in the palatability of food, we also determined the role of MOR in HFD-induced CPP. To address these issues, we used MOR knockout and their wildtype controls to investigate whether there is any difference in HFD consumption and HFD-induced reward between mice of the two genotypes.

Not only endogenous opioids, but also exogenous opioids may be involved in food intake and food reward. For instance, the local administration of a selective MOR agonist in the NAc, amygdala, hypothalamus, or VTA has been shown to increase palatable diet intake [16,17,18]. Conversely, a previous report has shown the lowering of food liking after treatment with a non-selective MOR antagonist [19]. Furthermore, MORs have a strong correlation with the intake of palatable diet intake [20]. Together, these findings suggest that there is a crosstalk between food reward and the rewarding actions of opioids. We hypothesized that prior exposure to HFD induces sensitization via the release of endogenous opioids acting at MORs. Thus, we also assessed if a prior conditioning with HFD would alter the rewarding action of oxycodone, a relatively selective MOR agonist.

Evidence exists to suggest a strong relationship between sex hormones and opioids. Interestingly, populations of neurons containing proopiomelanocortin (POMC, the beta-endorphin precursor) have been reported to express estrogen receptors (αERs). In addition, POMC mRNA fluctuates throughout the phases of the estrus cycle, and estrogen depletion via ovariectomy in females has been reported to decrease POMC mRNA [21]. Furthermore, knocking out the αER on POMC neurons leads to an increase in food consumption and body weight in female but not male mice [21]. Moreover, both MOR and αER are present in the arcuate nucleus of the hypothalamus (ARC) and NAc [1,22,23], raising the possibility that an interaction between these two receptor types is likely, and thus male/female differences may be present in food intake and food reward or in the involvement of MORs in food reward. Considering that the rewarding actions of drugs of abuse can be different between males and females [24], we also assessed if sex-related differences exist in HFD-induced reward, crosstalk between food reward and oxycodone reward, or in the involvement of MORs in these responses.

## 2. Materials and Methods

### 2.1. Subjects

Age-matched male (25–30 g) and female (20–21 g) C57BL/6J mice as well as male (30–32 g) and female (20–21 g) mice lacking MOR backcrossed for 12 generations on a C57BL/6J mouse strain bred in-house were used throughout. Each group consisted of six male and six female C57BL/6J mice or mice lacking MOR and their wildtype littermates. The original breeders of each line were purchased from Jackson Laboratories (Bar harbor, ME, USA). Subjects were kept one mouse per cage in a temperature-controlled (22 ± 3 °C) under a 12 h/12 h light/dark cycle (6 am light on and 6 pm light off). Animals had access to regular laboratory chow and water ad libitum except during the experiment. All the procedures were in accord with the NIH guideline for the use of animals in research and approved by the Institutional Animal Care and Use Committee (IACUC) at Western University of Health Sciences (Pomona, CA, USA).

### 2.2. Drug

Oxycodone hydrochloride, purchased from Sigma Aldrich (St. Louis, MO, USA), was dissolved in normal saline (0.9% sodium chloride solution) and injected intraperitoneally (i.p.) to each mouse during the conditioning at a dose of (5 mg/kg) per body weight.

### 2.3. Diets

High fat diet (HFD) for rodents was purchased from Research Diets (New Brunswick, NJ 08901, USA). The diet contains 60 Kcal % fat (Code name D12492, which contains 245 gm of fat generated from lard, 200 gm lactic protein and 125 gm carbohydrate in each 773.85 gm). The regular chow diet (RCD) for rodents contains 18% protein (protein 18.6%, fat 6.2%, and carbohydrate 44.2%) and was purchased from Teklad Global Diet (Madison, WI, USA).

### 2.4. To Determine If Binge Eating of HFD Induces CPP or Alter Oxycodone Reward and If Sex-Related Differences Exist in These Responses

We used an unbiased and counterbalanced place conditioning paradigm, widely used as an animal model of reward [25], to determine if a high fat diet would induce CPP or alter the rewarding effect of oxycodone. We used both male and female mice and assessed if sex-related differences exist in HFD-induced CPP. The place conditioning paradigm is the same as that of our earlier reports [26,27,28]. Briefly, in this paradigm, animals are conditioned with a drug or another agent in one of the conditioning chambers and with an inert substance, usually saline or another vehicle, and animals are tested for place preference toward the conditioning chambers before and after the conditioning. The conditioning chambers are distinguishable from each other by the inclusion of visual, olfactory and tactile cues. If the agent is rewarding, the animal will prefer the drug-paired chamber over the vehicle-paired chamber. In contrast, if the drug is aversive, the animal will avoid the drug-paired chamber. The place conditioning protocol consisted of three phases: (1) preconditioning, (2) conditioning, and (3) postconditioning. The preconditioning test was conducted on day 1 to assess the baseline place preference of each mouse toward the conditioning chambers. On this day, each mouse was placed in the gray central neutral chamber with both guillotine doors opened and allowed to explore the conditioning chambers for 15 min. The conditioning chambers were distinguishable from each other by the presence of visual cues, i.e., one of which had one-inch horizontal and the other one vertical black and white stripes. The amount of time that the mice spent in each chamber was recorded. Subjects who spent more than 67% or less than 33% of the total time (900 s) in any of the chambers were excluded from the remainder of the study, as per an earlier report [29]. Mice were then divided into two groups: (1) the HFD group and (2) the control (RCD) group. The HFD group received conditioning in the next four days, two with the HFD in one of the conditioning chambers or paired chamber (PCh) and two with the RCD in the opposite chamber or non-paired chamber (NPCh). The HFD was paired with the vertically striped chamber for some mice and with the horizontally striped chamber for other mice. We had an equal number of male and female mice assigned to each chamber. The control group received conditioning with RCD in both chambers, but one of the conditioning chambers was considered as the PCh and the other as NPCh. The conditioning was carried out from 3–5 pm (2 h) each day. On day 6, mice were then tested for place preference for 15 min (test 1), as described for day 1. Given that we did not observe any preference toward the PCh, we continued with two additional sets of conditioning and tested mice for placed preference. After the third test for place preference, they received conditioning for 16 h (6 pm–10 am) for the next four days. Mice were then tested for place preference. The rationale for overnight conditioning was that we hypothesized that a longer conditioning with HFD would induce reward.

We then tested if HFD conditioning for the rewarding action of oxycodone to assess if prior conditioning with HFD would alter the rewarding action of oxycodone, a relatively selective MOR agonist. To this end, the day after the last conditioning, mice were conditioned with oxycodone (5 mg/kg, i.p.) in the PCh and saline in the NPCh for one hour. The choice of the dose of oxycodone was according to an earlier study [30]. On the following day, mice received the alternate treatment and were conditioned to the opposite chamber. Twenty-four h later, mice were tested for a place preference toward the chambers, as described for day 1. Figure 1 illustrates a schematic presentation of the place conditioning protocol.

### 2.5. To Assess the Role of MOR in HFD-Induced Reward and If Sex-Related Differences Exist in These Responses

The experimental procedure was the same as described above, except MOR knockout and wildtype mice were used, and mice of both genotypes were exposed to HFD or RCD on alternative days. Given that we did not observe any place preference in mice of the RCD group, we did not include the control group in this experiment in order to reduce the number of mice used.

### 2.6. Data Analysis

The data are presented as means ± standard errors of the mean (±SEM) of the amount of time that mice spent in the paired chamber, and the amount of food (g) and calories (g) consumed in the conditioning chambers. All data were analyzed using a three-way analysis of variance (ANOVA) followed by the Fisher’s LSD *post hoc* test for multiple comparison. *p* < 0.05 was considered statistically significant. Each group/genotype contained six male and six to eight female mice.

## 3. Results

### 3.1. Conditioning with a HFD for 16 h but Not 2 h Induced CPP and Enhanced the Rewarding Action of Oxycodone in Male and Female C57BL/6J Mice

Figure 2 illustrates the amount of time that male (right panel) and female (left panel) mice spent in the paired chamber (PCh) in RCD and HFD groups. A mixed-effect ANOVA revealed a significant effect of diet (F (1, 204) = 38.93; *p* < 0.0001) and session (F (5, 204), =3.98; *p* < 0.01) and a trend for the effect of sex (F (1, 204) = 2.69; *p* = 0.10) but no interaction between the factors (*p* > 0.05). Subsequent analysis of data in male mice (Figure 2, left half of the panel) showed that STC with HFD failed to induce preference toward the PCh in these mice, as there was no significant difference between the baseline preference (BL) vs. Test 1, 2 or 3 (*p* > 0.05). On the other hand, when mice were exposed to LTC with HFD (labeled in the figure as ON), they spent more time in the PCh on the test day after LTC with HFD than the day of baseline (BL) test for preference toward the PCh (*p* < 0.05; Figure 2, ON vs. BL). In addition, there was a significant (*p* < 0.05) increase in the amount of time that mice of the HFD group spent in the PCh on this test day than mice of the RCD group (Figure 2). In contrast, mice in the RCD group did not exhibit any preference toward the PCh on any of the test day compared to the initial test for baseline preference toward this chamber (*p* > 0.5). Similarly, oxycodone (5 mg/kg) failed to induce preference toward the PCh in these mice (Figure 2, *p* > 0.05). In contrast, mice of the HFD group exhibited a significantly greater preference toward the PCh after oxycodone conditioning compared to baseline preference test day (Figure 2; *p* < 0.01). In addition, there was a significant increase in the amount of time that mice in the HFD group spent in the PCh on this day compared to mice in the RCD group (*p* < 0.01; Figure 2). 

While there was a significant preference toward the PCh in male mice after LTC conditioning with HFD and oxycodone conditioning, female mice failed to exhibit a preference toward the PCh on any test day vs. baseline preference test day (*p* > 0.05; Figure 2, right half of the panel), suggesting that some male/female differences exist in HFD-induced preference toward the PCh. Female mice of the HFD group spent significantly more time in the PCh than mice of the RCD group (*p* < 0.05). A trend toward greater preference toward the PCh following the LTC conditioning with HFD was observed in male vs. female mice (*p* = 0.05). Together, these results suggest that LTC with HFD induced conditioned place preference (CPP) in male but not female mice. We also observed a greater CPP response following oxycodone conditioning in both male and female mice of the HFD group compared to RCD group.

### 3.2. LTC but Not STC with HFD Induced CPP and Enhanced the Rewarding Action of Oxycodone in Wild-Type (MOR+/+) but Not MOR Knockout (MOR−/−) Mice

Figure 3 depicts the amount of time that male (left panel) and female (right panel) mice lacking MOR (open squares) and their wildtype controls (closed squares) of the RCD and HFD groups spent in the PCh. A mixed effect ANOVA revealed a significant effect of genotype (F (1, 60) = 41.48, *p* < 0.0001) and session and genotype interaction (F (5, 60) = 3.27, *p* < 0.05), but no significant effect of other factors or interaction between other factors (*p* > 0.05). Fisher’s LSD post hoc test showed that male wildtype mice spent more time in the PCh following the LTC with HFD (*p* < 0.05) and after oxycodone conditioning (*p* < 0.01) compared to the initial test for basal preference toward this chamber (Figure 3, the left panel, closed squares). These mice also exhibited greater preference compared to male knockout mice after LTC with HFD and oxycodone conditioning (*p* < 0.05). Although the baseline appears different between wildtype and knockout mice, it did not reach the level of significance (*p* = 0.09), showing that the difference in preference following LTC and oxycodone was not due to baseline differences between wildtype and knockout mice. As we observed in C57BL/6J mice, female wildtype mice failed to exhibit any CPP except following oxycodone conditioning, as evidenced by a significant increase in the amount of time that female C57BL/6J mice spent in the PCh on the test day following oxycodone conditioning compared to baseline test (Figure 3, right panel, closed squares). However, the difference between male and female mice was not present in these mice although there was a trend (*p* = 0.08) toward an increase in the amount of time that male mice spent in the PCh following LTC with HFD compared to female wildtype mice. While male mice lacking MOR did not exhibit place preference or place aversion after conditioning with HFD or oxycodone, female mice expressed aversion following oxycodone conditioning, i.e., there was a significant decrease in the amount of time that these mice spent in the PCh on the test day following oxycodone conditioning compared to the baseline test (*p* < 0.05).

### 3.3. Food Intake and Calorie Consumption between Male and Female C57BL/6J Mice of the RCD and HFD Groups

Food intake (upper panel) and calorie consumption (lower panel) in male and female mice is shown in Figure 4. Three-way repeated measures ANOVA revealed a significant effect of session (F (7, 192) = 255.1, *p* < 0.0001), sex (F (1, 192) = 10.44, *p* < 0.01) and diet (F (1, 192) = 274.40, *p* < 0.0001). There was also a significant interaction between diet and session (F (7, 192) = 7.36, *p* < 0.0001) but no other interactions were observed. The Fisher’s LSD post hoc test showed that the HFD intake was higher in males compared to females in the last session of the LTC with HFD (*p* < 0.0001; Figure 4; upper panel). There was also a robust trend toward an increase in HFD intake in male vs. female mice on the first session of the LTC with HFD (*p* = 0.06). Similarly, calorie intake was higher in the male mice of the HFD group (Figure 4, lower panel). Data analyses revealed a significant effect of session (F (7, 192) = 270.9, *p* < 0.0001), sex (F (1, 192) = 12.05, *p* < 0.001) and diet (F (1, 192) = 982.6, *p* < 0.0001). Fisher’s LSD post hoc test showed that there was an increase in caloric intake in male compared to female mice on the last two sessions of conditioning with HFD (Figure 4, bottom panel).

### 3.4. Comparison of Food Intake between Male and Female MOR Knockout vs. Wildtype Mice

Figure 5 depicts food consumption in male and female wildtype (upper panel) and knockout (lower panel) mice. A mixed effect ANOVA of data showed a significant effect of session (F (7, 176) = 201.8; *p* < 0.0001) and sex (F (1, 176) = 18.5; *p* < 0.0001), but no effect of genotype and the interactions between factors (*p* > 0.05). Subsequent analyses revealed a significant (*p* < 0.001) increase in HFD intake in male than female wildtype mice on day 17 (D17; Figure 5, top panel). There was also an increase in HFD intake in male wildtype than male knockout mice on this day (*p* < 0.01). For the calorie intake, a mixed effect ANOVA revealed a similar result, i.e., a significant (*p* < 0.001) increase in calorie intake in male compared to female wildtype mice as well as in male wildtype vs. male knockout mice on day 17 (D17; Figure 5, lower panel).

## 4. Discussion

The main findings of the current study are that male but not female mice exhibited a significant place preference only following long access to HFD in the conditioning chamber. The preference becomes more robust following conditioning with oxycodone, which failed to induce any significant preference in control mice, i.e., mice conditioned with a RCD in both conditioning chambers. These findings reveal that LTC with HFD induces CPP in male but potentiates oxycodone-induced reward in both male and female mice, as we only observed CPP following oxycodone in mice of the HFD but not the RCD group. The action of oxycodone was mediated via MOR, as we did not observe any CPP in MOR knockout mice of the HFD group. We also observed that male mice lacking MOR failed to exhibit any CPP following STC or LTC, suggesting that the rewarding action of HFD may involve MOR, but male/female differences may exist in this response.

Previous studies have shown that the overconsumption of fat-rich palatable diets induced drug-like reward hyposensitivity [31]. Importantly, exposure to HFD elicits changes in accumbal dopamine [32,33,34,35]. However, to the best of our knowledge, no prior study has evaluated whether HFD would induce reward. Therefore, using the place conditioning paradigm as a model of reward [25], we assessed if conditioning with HFD for 2 h, which may lead to binge eating in both male and female mice (Figure 4), could induce reward in C57BL/6J mice. Our data showed that short-term (2 h) conditioning with HFD failed to induce reward in male or female C57BL/6J mice. When male and female mice received 16-h conditioning with HFD, it elicited a significant CPP response in male but not female mice. We link this CPP response to the increased HFD consumption compared to 2 h conditioning, when animals had access for a short time to the HFD. There was a 2–3-fold increase in food intake during the LTC compared to STC. Although STC led to binge eating, it did not induce CPP in male or female mice. Thus, the duration of conditioning and the amount of HFD consumed may be essential to result in a significant preference toward the chamber paired with HFD. It is tempting to propose that a longer exposure time to the HFD may have increased dopamine in the NAc, leading to a significant CPP response.

An earlier study claimed that palatable foods recruit the endogenous opioid system [36]. Another study reported that intermittent sugar intake led to opioid dependence [37]. Considering that exposure to HFD elicits changes in accumbal dopamine [32,33,34,35], that the MOR agonists increase extracellular dopamine [38,39], and that the endogenous opioid system has been implicated in the palatability of food [16,40], we proposed that prior conditioning with HFD would enhance the rewarding action of oxycodone, a mu opioid receptor agonist [41]. To test this possibility, we used a single conditioning paradigm and a low oxycodone dose (5 mg/kg, i.p.) which did not induce CPP in male or female control mice (RCD group) in the current study. In contrast, both male and female mice conditioned with HFD in one conditioning chamber and RCD in the opposite chamber (HFD group) displayed a robust CPP response following the single conditioning with oxycodone when paired with the HFD-paired chamber (PCh). Our results are consistent with previous reports [42,43,44] suggesting that prior exposure to HFD enhances the rewarding action of addictive drugs; in this case, oxycodone. However, given that it is generally thought that palatable food is rewarding (for a review, see [45]) and that mice already showed CPP following LTC with HFD, it is unclear whether conditioning with HFD potentiated the oxycodone reward or the enhanced reward following oxycodone was a result of the CPP induced by HFD. Thus, future studies are needed to delineate between these two possibilities.

Chronic HFD exposure has been reported to alter MOR gene expression differentially between male and female mice [29,46]. Previous research has reported that fat-rich palatable foods are rewarding [45] and may cause the release of endogenous opioids known to govern palatable food intake [47]. Thus, we hypothesized that HFD induces reward, and that MOR is involved in this response. We used MOR knockout mice and their wildtype littermates to test our hypothesis. We also determined if any sex-related difference exists in the CPP response induced by HFD or the potentiation of oxycodone reward. We observed a significant CPP response in male but not female wildtype mice after LTC with HFD and following oxycodone conditioning. This response was blunted in male mice lacking MOR, suggesting that MOR may be involved in HFD-induced reward in male mice. Despite this, the food intake was the same between mice of the two genotypes except on the last conditioning day (day 17). Thus, the lack of CPP in knockout mice was not due to a decrease in food intake, as they were consuming an equal amount of HFD compared their wildtype littermates. Together, these findings suggest that MOR is necessary for the expression of CPP induced by LTC with HFD and oxycodone in mice, but sexual dimorphism exists in this response, which requires further investigation. For example, it would be necessary to determine the interaction between sex hormones and MOR, as its expression can be altered during the phases of the estrous cycle [48]. The NAc can be a potential target to assess for the interaction between MOR and sex hormones, as both MOR and αER are present in the NAc, and this brain area is implicated in food reward and drug reward [1,22,23].

The results of the current study should be interpreted with caution, because our study was underpowered, especially in female mice, given that we did not record phases of the estrous cycle in these mice. Given that there was no significant difference between the amount of time that C57BL/6J and MOR wildtype mice spent in the PCh (Appendix A), we combined the results of these mice with their respective sex group, which led to an unequal sample size between the HFD group and the RCD group as well as between wildtype mice and knockout mice, and may have confounded the interpretation of the data.

## 5. Conclusions

This is the first report to reveal that 16 h but not 2 h conditioning with HFD induces reward in male but not female mice but enhances the rewarding action of oxycodone in both male and female C57BL/6J mice as well as in MOR wildtype mice. These responses were blunted in male MOR knockout mice, suggesting that a sexual dimorphism in HFD-induced reward in the absence of MOR may exist. However, further studies are needed to delineate the underlying mechanism of the crosstalk between HFD and oxycodone reward and sexual dimorphism in these responses. Understanding the neurobiology of HFD-induced reward may help in the design and development of therapy for binge eating and other eating disorders.

## Figures and Tables

**Figure 1 life-13-00619-f001:**
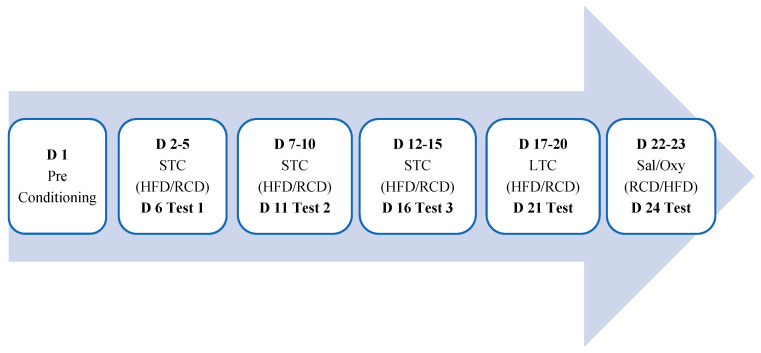
A schematic presentation of the experimental procedure in mice. **Abbreviations: D**: day; **STC**: short-term conditioning; **LTC**: long-time conditioning; **OXY**: test after oxycodone conditioning; **HFD**: high fat diet; **RCD**: regular chow diet; **Sal**: saline, **Oxy**: oxycodone.

**Figure 2 life-13-00619-f002:**
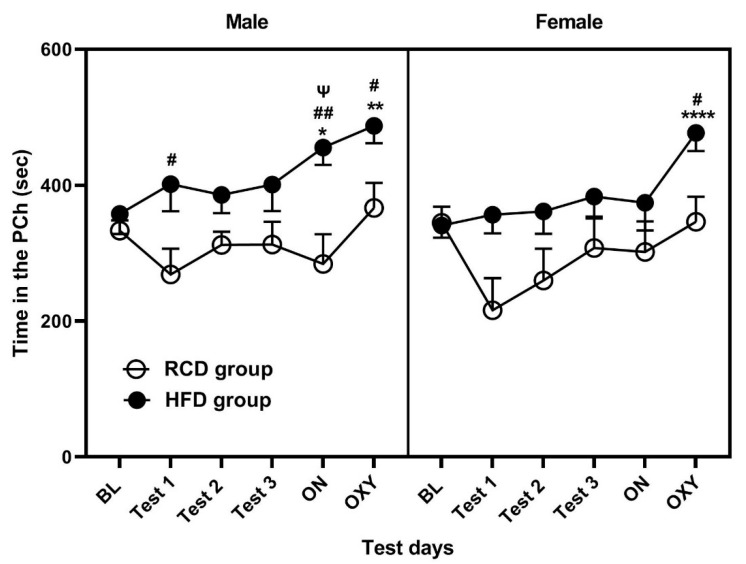
Preference toward the paired chamber (PCh) in male (left panel) and female (right panel) mice of the RCD and HFD groups. Data represent mean ± standard error of the mean (±S.E.M) of the amount of time that mice of each group and sex spent in the PCh on the preconditioning (BL) and postconditioning (Tests 1, 2, 3, ON and OXY) days. * *p* < 0.05, ** *p* < 0.01 and **** *p* < 0.0001 indicate a significant difference on the ON and OXY test days vs. BL in male mice of the HFD group; # *p* < 0.05 and ## *p* < 0.01 indicate a significant difference in the amount of time that male or female mice of the HFD group spent in the PCh vs. mice of the RCD group. ^Ψ^
*p* = 0.05, a robust trend toward higher preference in male vs. female mice of the HFD group.

**Figure 3 life-13-00619-f003:**
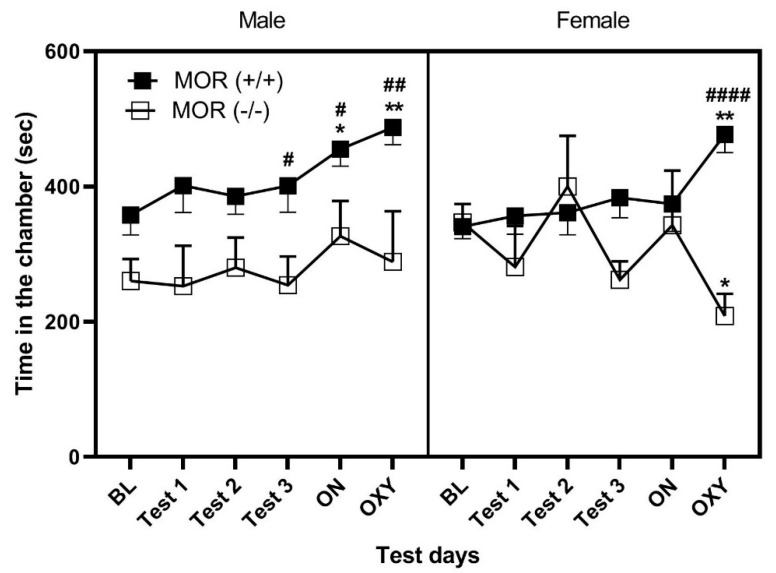
Place preference in male (**left panel**) and female (**right panel**) mice lacking MOR (open squares) and their wildtype controls (closed squares) following conditioning with HFD in one chamber (PCh) and RCD in the opposite chamber (NPCh) followed by oxycodone in the PCh and saline in the NPCh. Data represent the mean ± standard error of the mean (±S.E.M) of the amount of time that mice spent in the PCh in mice lacking MOR (−/−) (*n* = 6 mice per sex) and their wildtype controls (*n* = 12–14 mice per sex). * *p* < 0.05, ** *p* < 0.01, a significant increase or decrease in the amount of time that wildtype or knockout mice spent in the PCh after overnight conditioning (ON) with HFD or oxycodone conditioning vs. their respective baseline (BL); # *p* < 0.05, ## *p* < 0.01 #### *p* < 0.0001, a significant difference in the amount of time that wildtype mice spent in the PCh vs. their respective knockout mice.

**Figure 4 life-13-00619-f004:**
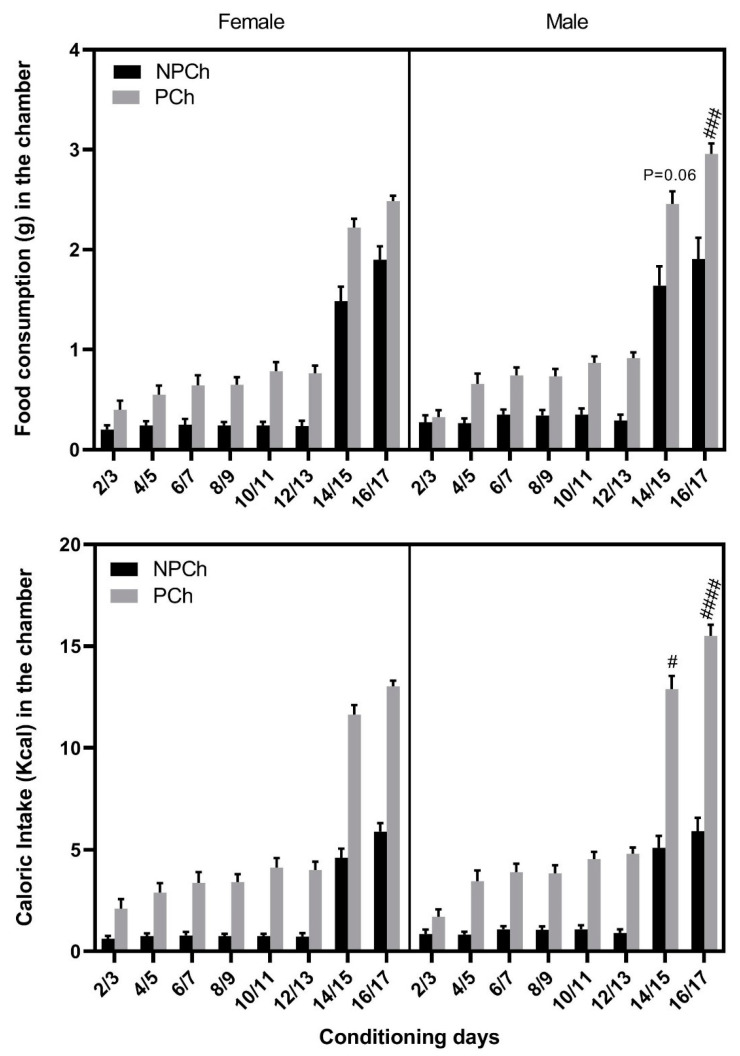
Comparison of food and caloric intake between female and male mice. Data represents mean ± standard error of the mean (±S.E.M) of food intake (**upper panel**) and calorie intake (**lower panel**) in the PCh vs. NPCh in female (*n* = 6) vs. male (*n* = 6) mice of the HFD group. # *p* < 0.05, ### *p* <0.001 and #### *p* < 0.0001 indicate a greater HFD consumption or caloric intake in male than female mice.

**Figure 5 life-13-00619-f005:**
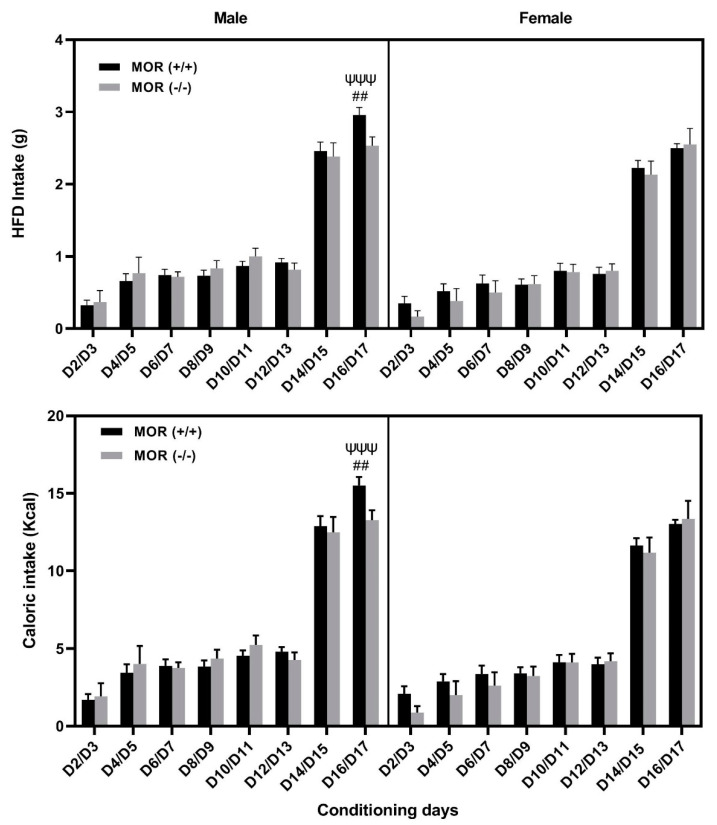
HFD consumption (**upper panel**) and calorie intake (**lower panel**) in male and female wildtype and knockout mice. Data represent mean ± standard error of the mean (±S.E.M) of HFD intake in male and female mice lacking MOR (*n* = 6 of each sex) and their wild-type littermates (*n* = 12 mice of each sex). ## *p* < 0.01, a significant increase in food intake on male wildtype vs. male knockout mice; ^ΨΨΨ^
*p* < 0.001, a significant increase in male vs. female wildtype mice.

## Data Availability

Data are stored in a OneDrive folder that belongs to Western University and will be made available upon reasonable request.

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
