# Peer review of "The Role of Mu Opioid Receptors in High Fat Diet-Induced Reward and Potentiation of the Rewarding Effect of Oxycodoneâ€"

_life, 2023, doi:10.3390/life13030619_

Round 1
Reviewer 1 Report
Comments to the authors:
The present study, “the role of mu-opioid receptors in high fat diet-induced reward and potentiated of the rewarding effect of oxycodone”, examined the involvement of mu-opioid receptors in high fat diet-induced rewarding effects and whether sexual differences in this effect; moreover, whether mu-opioid agonist oxycodone would increase the high fat diet-induced rewarding effect. The present study showed that high-fat diet-induced conditioned place preference in males and females following long-term conditioning (LTC). This response was facilitated by oxycodone. The response was mediated by mu-opioid receptor (MOR). However, high fat diet-induced conditioned reward occurred in the LTC but not short-term conditioning only in female but not male MOR knockout mice. The study is so interesting. However, some deficits should be concerned.
Major points:
1. The present study involved some crucial issues. First, why could the high-fat diet induce the rewarding effect? Nevertheless, the present manuscript has cited one paper and described that “…excessive high-fat diet consumption has been implicated in dysregulated dopamine and opioid gene expression [7]...”. It should be added to previous evidence, and it needs to be clarified. Second, What kind of neural substrates were involved in high-fat diet-induced conditioned place preference? This point reminas uncelar. It should be clarified further.
2. The data of figure 2 should be analyzed by 2 x 2 x6 three-way (treat vs. pair vs. sessions) mixed ANOVA firstly. It should identify which factors were significant differences and then omit the factor that was nonsignificant differences. Therefore, the author should report the whole 2 x 2 x6 three-way (treat vs. pair vs. sessions) mixed ANOVA. Later, the authors analyzed two-way ANOVA and/or one-way ANOVA. The analysis strategy is structuralized ways. However, the analysis method does not pick up partial factors that the authors want and thereby analyze them. Remember that when the main effect of the factor and the interaction of factor 1 and factor 2 were significant differences, these factors were further analyzed. Otherwise, two-way ANOVA or one-way ANOVA cannot further analyze these nonsignificantly different factors.
3. Figure 3 also had the same statistical problems. The authors should first use 2 x 2 x 6 mixed three-way (MOR vs. pair vs. sessions) ANOVA to analyze the time in the chamber. This point should be clarified.
4. The data in Figure 4 should be first analyzed by 2 x 2 x 2 x 6 (sex vs treat vs pair vs session) mixed four-way ANOVA. Because the authors manipulate numerous variances, including sex, treat, pair, and sessions, they need to analyze the data by four-way ANOVA to rule out some factors related to nonsignificant differences.
5. The data of Figure 5 should be analyzed by 2 x 2 x 2 x 6 (MOR vs sex vs pair vs session) mixed four-way ANOVA first. Then, rule out the factors of nonsignificant differences and analyze the data using three-way, two-way, or one-way ANOVA.
6. Why did oxycodone potentiate high-fat diet-induced conditioned reward? It should be clarified. Furthermore, how did the mu-opioid agonist oxycodone administrations interact with the dopamine system and thereby influence high-diet fat-induced conditioned reward?
7. How did the present results provide some clinical implications or contributions?
8. What are the experimental limitations? It should be written in the Discussion section.
In summary, the study needs to be majorly revised. The current status of the manuscript cannot be considered for acceptance.

Reviewer 2 Report
Major comments:
There are some issues regarding the design and presentation of their CPP protocol. First, it was not mentioned whether each compartment in the CPP apparatus had distinct features from each other (such as one having smooth flooring, the other having textured flooring). Having these features are essential when performing CPP for mice to clearly associate “treatment-paired” and “non-treatment paired” chambers. It was also unclear whether the HFD-paired chamber was linked to only a single type of chamber all throughout the experiment. How did the authors designate the treatment with the chamber, and a mouse to a chamber? Randomly or based on less preferred chamber? Also, was the stay duration in the gray chamber excluded in all data? Second, their basis for place preference (comparing the difference between PCh and NPCh during post-conditioning) may be weak. Studies usually compare the stay duration in the treatment-paired chamber between post- and pre-conditioning as an indication of changes in place preference. In their results (Figure 2a and 3a), they deemed the behavior of mice during ON in treated female mice and female MOR (-/-) mice as expressions of increased place preference. However, when looking at their PCh duration in ON and BL, the duration in PCh seemed to remain unchanged; it is just that the stay duration in NPCh decreased. In that case, it might be uncertain whether this is true CPP behavior. Furthermore, looking at these results seemed that the total duration in both chambers is just around 600 s. If this means just an increase in the stay duration in the gray area (up to 300 s), it does not contribute to CPP behavior. Similarly, when comparing the duration in the PCh between control and treated mice, there might be no significant differences, rendering the presumed expression of CPP questionable. Thus, it may simplify and improve the clarity and integrity of their results if they compare the stay duration of treated vs. control (or MOR (+/+) vs. MOR (-/-)) mice in the PCh only.
Minor comments:
1. “ARC” was mentioned in the introduction section but was not defined.
2. Introduction section (last paragraph) contained several text errors (contining, repeating “the”, beta-edorphin, etc.).
3. Authors should cite the reference for selecting 5 mg/kg dose of oxycodone.
4. Materials and Methods section 2.4 is too long for a subheading. Can be rephrased to “HFD CPP and its involvement, along with sex differences, in oxycodone reward”.
5. Refrain from repeating “a mu opioid receptor agonist” every time oxycodone is mentioned.
6. “Treated group” may cause some confusion, maybe “HFD-exposed group” is more appropriate.
7. “LAC” and “SAC” were not defined.
8. Results section 3.2, “…no CPP during test 1 but during test 2 but not test 3 they showed a modest CPP response…” can be changed to “…no CPP during test 1 and 3, but in test 2, they showed modest CPP response…”.
9. Treated female C57BL/6J mice and treated female MOR (+/+) mice would presumably exhibit the same response in these experiments. How would the authors explain the lack of CPP in Test 3 in Figure 2a, right panel, and the presence of CPP in Test 3 in Figure 3a, left panel?
Reviewer 3 Report
"The role of mu opioid receptors in high fat diet-induced reward and potentiation of the rewarding effect of oxycodone" by Iqbal and colleagues describes an experiment connecting HFD with exogenous oxycodone intake via the MOR pathway. I must say I did not find any major reason for criticism: the study is sound in all scientific aspects related to behavioral and biological sciences. I have a few minor points to raise, below:
- The extensive usage of acronyms makes the paper hard to read sometimes: I would suggest introducing a table with all the 10+ acronyms used recurringly. Also, the acronyms should be explained the first time they are introduced in the main text. For example, MOR is introduced in page 2, around line 15, but the acronym is explained one paragraph 19 lines later. Finally, some concepts that may not be familiare with the readers of Life should be better introduced, like the design and purpose of CPP.
- How do the authors explain the differences between males and females from a genetic and molecular point of view? This should be extensively discussed.
- Figure 2 should clearly state "female" and "male" in the titles of the plots. Also, given the symmetry of SEM bars and the overlap between PCh and NPCh error bars, I would use the top bar for PCh, and the bottom bar for NPCh (or vice versa).
- Figure 3, minor point: should have both panels aligned.
- Figure 6: the meaning of three "#" is not described in the legend (I assume it is p<0.001). Also, both here and in the next Figure, the changes in female mice seem borderline significant, and the lack of a low p-value are certainly due to the low sample size. The authors may introduce the symbol for a less stringent p-value threshold, e.g. p<0.1.
"The role of mu opioid receptors in high fat diet-induced reward and potentiation of the rewarding effect of oxycodone" by Iqbal and colleagues describes an experiment connecting HFD with exogenous oxycodone intake via the MOR pathway. I must say I did not find any major reason for criticism: the study is sound in all scientific aspects related to behavioral and biological sciences. I have a few minor points to raise, below:
- The extensive usage of acronyms makes the paper hard to read sometimes: I would suggest introducing a table with all the 10+ acronyms used recurringly. Also, the acronyms should be explained the first time they are introduced in the main text. For example, MOR is introduced in page 2, around line 15, but the acronym is explained one paragraph 19 lines later. Finally, some concepts that may not be familiare with the readers of Life should be better introduced, like the design and purpose of CPP.
- How do the authors explain the differences between males and females from a genetic and molecular point of view? This should be extensively discussed.
- Figure 2 should clearly state "female" and "male" in the titles of the plots. Also, given the symmetry of SEM bars and the overlap between PCh and NPCh error bars, I would use the top bar for PCh, and the bottom bar for NPCh (or vice versa).
- Figure 3, minor point: should have both panels aligned.
- Figure 6: the meaning of three "#" is not described in the legend (I assume it is p<0.001). Also, both here and in the next Figure, the changes in female mice seem borderline significant, and the lack of a low p-value are certainly due to the low sample size. The authors may introduce the symbol for a less stringent p-value threshold, e.g. p<0.1.
Round 2
Reviewer 1 Report
Dear Authors,
Greetings! The authors have fully responded to all comments.
The current status of the manuscript can be considered for acceptance. Thank you.
Andrew
Author Response
We appreciate your consideration and we thank you for your time and dedication.
Sincerely,
Kabirullah Lutfy, Ph.D.
Reviewer 2 Report
I commend the authors for considering the comments and revising their manuscript accordingly, improving the readability and quality of their paper. There are some queries remaining, that once answered, may render their work possible for publication.
In Figure 3, there is a slight difference between male MOR (+/+) and MOR (-/-) mice during BL. Please provide the statistical analysis between male MOR (+/+) and MOR (-/-) mice during BL, just to verify the lack of significant difference (need not to be indicated in the revision). However, if there is a significant difference, there might be an inherent bias among male mice regarding chamber preference.
Author Response
Dear Reviewer,
Thank you so much for your consideration and time. As we reported in the original manuscript, there was no difference in the amount of time that mice spent in the paired chamber (PCh) vs. the non-paired chamber (NPCh). In the revised version, we presented only the data in the PCh. Although there is a reduction in male MOR knockout than wildtype mice, it was not statistically significant. Considering that there is an increase in the LTC test day and oxycodone CPP test day compared to baseline in wildtype mice, we do not believe that the initial difference between the two genotypes could be the reason for the preference in wildtype mice.
Please let us know if we need to make any other changes.
Sincerely,
Kabirullah Lutfy, Ph.D.